# Development of a Predictive Model for Runway Water Film Depth

**DOI:** 10.3390/s25072202

**Published:** 2025-03-31

**Authors:** Peida Lin, Chiapei Chou

**Affiliations:** 1Department of Civil Engineering, National Taiwan University, Taipei 106, Taiwan; cpchou100@gmail.com; 2Taiwan Transportation Safety Board, New Taipei City 231, Taiwan

**Keywords:** water film depth, empirical prediction model, hydroplaning, runway transverse slope, multiple linear regression, mean texture depth

## Abstract

Water film depth (WFD) on runways is a key factor contributing to aircraft hydroplaning during takeoff and landing. Thus, the early measurement or prediction of WFD during rain is critical for reducing accidents. Most existing WFD prediction models are derived from experiments conducted on road surfaces. However, an accurate prediction of WFD on runways and reduced hydroplaning risk require a precise empirical prediction model. This study conducted experiments involving four parameters—rainfall intensity, pavement mean texture depth, drainage length, and transverse slope—to develop a WFD dataset specific to different runway conditions. The multiple linear regression method is employed to establish a model for WFD predictions. The proposed National Taiwan University (NTU) model’s predictability is compared with three existing empirical models using NTU and Gallaway datasets. The results clearly demonstrate the superior accuracy and robustness of the NTU model compared to the other evaluated models. The NTU model offers a precise and practical predictive formula, making it highly suitable for integration into contaminated runway warning and management systems. This study employed a laser displacement sensor and a programmable logic controller to obtain high-accuracy, high-sampling-rate WFD data. Modern automated data acquisition enables simultaneous measurement at multiple points and captures the complete WFD curve from zero to a stable depth, which was previously difficult to obtain.

## 1. Introduction

Statistics show that 29% of aircraft accidents occur during landing or takeoff on wet runways [1]. These accidents are mainly caused by rainfall accumulation, which creates a thick water film on the pavement surface. When aircraft land on wet runways, the water film significantly reduces friction [2,3], increasing the likelihood of hydroplaning or aquaplaning. This phenomenon heightens the risk of runway overruns, potentially resulting in substantial damage to the aircraft or passenger injuries and fatalities [4]. Consequently, measuring the WFD to assess runway conditions is important for ensuring the safety of aircraft operations.

The International Civil Aviation Organization (ICAO) Annex 14 [5] mandates the real-time reporting of standing water depths exceeding 3 mm on runways. This requirement, specified in the ICAO Global Reporting Format (GRF) for Runway Surface Conditions, has been in effect since November 2021 to reduce the risk of aircraft aquaplaning. In practice, measuring WFD on runways for landing or departing aircraft during adverse weather conditions is costly and hazardous for the staff involved. WFD measurement at specific pavement points using instruments is constrained by runway clearance requirements, making on-pavement installation challenging. The distance of instrument placement and adverse weather conditions also limit measurement accuracy. As a result, predictive models for predicting potential WFD in touchdown zones have gained increasing attention as an alternative.

Predictive models can be classified into two types: empirical and analytical models [1]. Empirical models are developed using experimental data to formulate mathematical expressions, while analytical models are based on analyses of physical and mechanical principles. Over the past decades, research has focused largely on developing WFD predictive models. Prediction accuracy depends heavily on the suitability of the selected parameters regardless of model type. Most predictive models incorporate independent parameters, such as pavement slope, rainfall intensity, drainage length, and mean texture depth. In general, drainage length and rainfall intensity are positively correlated with WFD, whereas pavement slope and mean texture depth are negatively correlated.

Analytical models require validation through experimental methods, but the reliability of these experiments depends on a well-designed framework and precise data acquisition. Ensuring the quality of the predictive model necessitates thorough planning and execution. In this study, optimal sensors will be selected to measure WFD, and data will be collected through experimental methods. The gathered dataset will then be used to develop a predictive formula for WFD.

## 2. Literature Review

### 2.1. Slippery Pavements

Thick water film on a slippery pavement reduces tire–pavement contact while increasing hydrodynamic forces, leading to a significant decrease in skid resistance. Therefore, accurate determination of the water film depth on airport runways is essential for predicting the additional landing roll distance of aircraft and assessing the potential risk of hydroplaning.

Toraldo et al. [6] found that the repeated use of runways can lead to rutting damage, and water accumulation in these ruts during heavy rainfall can greatly extend aircraft braking distances. Similarly, Zhu et al. [2] observed that when an aircraft lands on a runway with an average WFD of 8 mm, its braking time is approximately 1.6 times longer, and its braking distance is about 5.3 times greater compared to landing on a dry runway. Jiang and Wang [7] calculated friction coefficients at various speeds, which were used to predict hydroplaning speeds and estimate aircraft tire braking distances. Ling et al. [1] conducted a comprehensive assessment of factors influencing road and airport runway safety, concluding that water film depth is the primary factor contributing to loss of control and reduced friction.

Deng et al. [8] used multiscale power spectrum analysis of 3D surface textures to predict asphalt pavement friction, emphasizing WFD’s critical role in reducing tire–pavement friction. Gerthoffert et al. [3] employed a brush-based modeling approach to develop runway friction evaluation devices and showed that the likelihood of accidents on wet pavements is 10 times higher than on dry pavements.

### 2.2. Prediction Models for Water Film Depth

Luo et al. [9] and Ling et al. [1] reviewed prediction models for pavement drainage and categorized them into two types: empirical and theoretical models. For empirical models, Gallaway et al. [10] developed a large-scale indoor rainfall simulation apparatus and derived a mathematical formula as an empirical evaluation model based on experimental measurements. This model remains widely used for predicting and investigating water accumulation on airport runways. Chesterton et al. [11] adapted Gallaway’s formula to account for the unique characteristics of New Zealand’s ALPURT B2 highway. Pourhassan et al. [12] conducted laboratory experiments on a 2.4 m × 1.2 m chip-seal specimen, measuring 1784 WFD readings using a digital flowmeter and controlled rainfall intensities. Researchers concluded that WFD prediction models by Gallaway and David et al. [13] were less accurate for high-texture pavements ranging from 1.27 to 5.08 mm. Hence, they proposed two new empirical prediction models. Similarly, Guo et al. [14] constructed a 4 m × 6 m runway pavement and used nozzles, rain gauges, and water level gauges to simulate WFD on airport runways under the influence of heavy rainfall and wind. They further developed nonlinear regression formulas to describe the factors affecting WFD.

Gallaway’s experiments, which were conducted under low rainfall intensity, resulted in lower WFD measurements. In contrast, Pourhassan et al. [12] designed their experiment for high-intensity rainfall conditions. The predictive formulas developed in their study incorporated aggregate rubber content as an additional parameter, making them more suitable for predicting WFD on high-texture pavements.

Theoretical models take a different approach. Huebner et al. [13] developed a universal model through theoretical analysis to calculate the WFD distribution along the drainage paths on asphalt concrete pavements. Luo et al. [9] introduced an analytical model capable of dynamically simulating water flow on asphalt pavements during heavy rainfall. This model demonstrated higher accuracy compared to Gallaway’s models.

Snow and ice also significantly reduce tire–pavement friction, affecting deceleration and directional control. Midtfjord et al. [15] proposed a machine learning framework for real-time runway condition assessments. Using the XGBoost algorithm, they developed a comprehensive runway evaluation system to identify slippery conditions and predict slipperiness levels through regression models. The results showed that machine learning techniques can accurately model complex physical phenomena. Schulz et al. [16] applied fluid–pavement interaction principles to derive equations describing the hydrodynamics of water film, providing a theoretical basis for understanding WFD.

### 2.3. Parameters Affecting WFD

Selecting appropriate parameters is essential for developing predictive models to assess the risk of aircraft hydroplaning. Most WFD prediction models commonly select four parameters: pavement slope, rainfall intensity, drainage length, and mean texture depth, discussed as follows.

Xiao et al. [17] examined water flow characteristics and factors influencing runoff on asphalt pavements. Their findings indicated that WFD on runways is significantly affected by rainfall intensity, wind conditions, and pavement surface characteristics. Similarly, Alber et al. [18] evaluated the effects of road geometry (transverse and longitudinal slopes), pavement surface conditions (mean texture depth and rut deformation), and rainfall characteristics (intensity and duration) on the drainage capacity of rutted pavements. They observed strong interrelations and trends among these parameters.

Luo et al. [19] focused on evaluating surface drainage on asphalt pavements, highlighting the critical role of texture depth. They found that an increase in texture depth storage capacity reduces the water depth retained on the pavement surface. Kumar and Gupta [20] also reviewed factors influencing skid resistance between tires and pavements and concluded that pavement texture depth and rainfall intensity are key determinants of skid resistance. Li et al. [21] developed an empirical evaluation model considering rainfall intensity, slope, and drainage length. While their model proved suitable for runways, it did not account for the impact of texture depth on WFD.

Pavement slope is another crucial parameter. Pourhassan et al. [12] analyzed the relationship between WFD and slope, finding that increased slope enhances runoff due to gravitational forces. Their study confirmed a negative correlation between slope and WFD. Drainage length also significantly influences WFD on pavement surfaces; as the area exposed to rainfall and runoff volume grows, drainage length also increases. However, as runoff energy losses increase, flow rates decrease, resulting in more water being retained on the pavement surface. Thus, a positive correlation exists between drainage length and WFD.

### 2.4. Measurement Instruments for WFD

Xiao et al. [22] identified a limitation in previous studies, namely that most experiments require manual operation and visual observation of experimental data during testing. Operational errors in the instruments and discrepancies in reading measurements can affect the accuracy of WDF data. Accurate measurement of parameter values is essential for ensuring the quality of the prediction results. Therefore, reliable and appropriate measuring instruments are necessary to improve the accuracy and reliability of the prediction model. On asphalt pavements, contact measurement methods are less flexible and less precise compared to non-contact methods.

For non-contact measurement, Wu et al. [23] used a 1310 nm to 1510 nm laser to measure WFD. The measurement error was less than 0.1 mm for a measurement distance of 3 to 5 m and WFD under 9 mm. Yang et al. [24] employed 3D pavement data obtained through LiDAR combined with numerical methods to estimate the distribution of water film. Compared to traditional techniques, this method can quickly predict water film depth on pavements up to 10 m wide and 100 m long within 10 s. The results showed that, in rutting areas, the minimum hydroplaning speed could be lower than 90 km/h, significantly affecting driving safety. Ling et al. [1] reviewed and compared various sensors based on previous research and found that laser technology is generally superior due to its applicability in any environment and rapid response time. The operational and maintenance advantages of laser technology make it preferable over other measurement instruments, making its use for accurate measurements increasingly popular.

The sensors measure WFD using different principles, including Laser-Based Sensors, Ultrasonic Sensors, and Camera-Based Systems, which can generally be categorized as contact and non-contact types. The author has referenced the sensor data compiled by Ling [1] for WFD measurement and incorporated the two sensors used in this study, along with the NASA’s water depth gauge, into Table 1.

Ensuring high-quality experimental design and measurement data is crucial to developing an accurate WFD prediction model. Therefore, experimental equipment should provide uniform rainfall distribution across various points on the pavement, and suitable sensors must be selected for dynamic WFD measurements. This study constructs an experimental framework with these goals in mind and proposes a predictive model for WFD based on the findings.

## 3. Methodology

This study involved experimental setup and environmental construction, data measurement and model evaluation, as shown in Figure 1. The environmental setup included the installation of a water supply system, the construction of runway pavement, and the placement of sensors. A water supply system was used to simulate rainfall conditions, while an asphalt concrete pavement was prepared to replicate runways with different slopes and mean texture depths. Laser displacement sensors were installed to measure WFD dynamically.

Regarding the WFD prediction model, an empirical model proposed by Gallaway et al. has already been introduced. Additionally, various models with different levels of accuracy and reliability evaluation criteria have been adopted. Since using a combination of different comparison and evaluation approaches presents a significant challenge, this study focuses on exploring and analyzing commonly used empirical models and evaluation criteria.

Three evaluation metrics were employed, mean absolute error (MAE), relative absolute error (RAE), and relative root squared error (RRSE), to compare the predictive performance of the four existing WFD models with the model developed in this study.

### 3.1. Experimental Equipment

#### 3.1.1. Water Supply System

As shown in Figure 2, the layout of the water supply system was designed to simulate rainfall on pavement surfaces. This system consists of the following primary components: a water tank, water pump, water filter, steel drum, air compressor, water flow valve, flow meter, distribution device, water supply tubes, spray nozzles, and a drainage pipe. The water tank served as the reservoir, providing the necessary volume for rainfall simulation. Water was pumped from the water tank, filtered through a filtration device to remove impurities that could clog the nozzles, and directed into a high-pressure cylindrical steel drum. The air compressor, paired with a pressure regulator, a water flow valve, and a flow meter, maintained optimal water pressure to ensure steady flow to the distribution system. The water was then directed by the distribution device, which was mounted on a support frame, to the spray nozzles. Finally, excess water was collected through drainage pipes and returned to the water tank, allowing for efficient recycling within a closed-loop system.

#### 3.1.2. Pavement System

Figure 3 illustrates the design of the pavement system for the runway. Given the symmetrical nature of runways, the pavement system in this experiment was designed to represent only one side of the actual runway structure along its centerline. The upstream section of the pavement corresponds to the runway centerline, while the downstream section represents the area near the runway edge. The key independent parameters, which include the transverse width of the pavement, transverse slope, and mean texture depth, varied in accordance with the experimental design to achieve the desired configurations.

#### 3.1.3. Water Depth Measurement System

This study utilized specialized laser displacement sensors based on reflective principles to measure the WFD on the pavement with high accuracy. The laser beam from the sensor is projected at a fixed angle onto the surfaces of the water film and the pavement. The reflected light is collected by a receiver lens and directed to a sensor component. When the laser beam passes through the water medium, its propagation speed is slower than when passing through air, resulting in a measurable delay in the reflection time. This time delay varies with the depth of the water, enabling the prediction of water depth based on this principle. The configuration of the water depth measurement system is illustrated in Figure 4. This study employed a probe-type water level gauge (as shown in Figure 5) to measure the actual water depth, which corresponds to the depth measured by the laser displacement sensor.

Before the experiment, the probe-type water level gauge and laser displacement sensor were used to measure water depth, obtaining the correlation, Actual water depth = 2.9852 * Laser Displacement Sensor measured water depth, as shown in Figure 6.

### 3.2. Data Processing

Data acquisition involves two primary steps, measurement and processing, to obtain the final analytical dataset. In this study, key parameters, such as mean texture depth, pavement slope, rainfall intensity, and drainage length, were controlled. WFD measurements were recorded throughout the experiment. Given that WFD measurements are inherently unstable due to disturbances from raindrop splashes and water flow fluctuations, the simple moving average (SMA) method was applied to smooth the data. The average results represent the WFD corresponding to specific experimental parameters throughout the measurement.

Equations (1) and (2) outline the data acquisition process for generating stable measurement results. During an interval P, at the *i*-th iteration, *W_f_* denotes the WFD detected at time point *f*, where *f* ∈ P. At a given time point *x*,$\;f_{i}\left(x\right)$ represents the average WFD measured over the interval. These procedures are repeated to calculate a moving average as the temporary WFD value for each time point *x*. The iterations are performed n times, with *f*(*x*) denoting the average of all $f_{i}\left(x\right)$ iterations. The final *f*(*x*) value represents the WFD under the experimental design, which includes parameters such as average texture depth, pavement transverse slope, rainfall intensity, and drainage length.(1)$$f_{i}\left(x\right)=\{\begin{array}{c} \frac{\sum_{f=1}^{f=x}W_{f}}{x},\;\;x\leq b\\ \frac{\sum_{f=\left(x-b+1\right)}^{f=x}W_{f}}{b},\;\;x>b \end{array}$$(2)$$f\left(x\right)=\frac{\sum_{i=1}^{n}f_{i}\left(x\right)}{n}$$
where *x* = the detected time point, *i* = the *i*th iteration for sample point *x*, *b* = the moving period, *W_f_* = the detected WFD at time points of *f*, $f_{i}\left(x\right)$ = the mean value of *i*th iteration, P = the total period, *n* = the measuring times in P, and *f*(*x*) = the mean of all iterations of $f_{i}\left(x\right)$.

### 3.3. Model Evaluation

After conducting experiments under various scenarios outlined in the experimental design, this study will develop a predictive model for WFD and compare its performance against existing models proposed in prior research. The existing models include the Pourhassan model (Pourhassan et al. [12]), the Gallaway model (Gallaway et al. [25]), and the Empirical PAVDRN model (David et al. [13]), as presented in Equations (3)–(5). All of these models incorporate parameters common to this study, including rainfall intensity (I, in./h), mean texture depth (MTD, in), water path length (L, ft), and runway transverse slope (S, ft/ft). The WFD was measured using a gauge down to the pavement surface. Consequently, the measured WFD value was obtained by subtracting the MTD from the depth recorded by the gauge. As a result, the WFD value at the reference plane of the mean texture depth is zero.

Pourhassan model:(3)$$WFD=0.00639\times (MTD^{-0.46}L^{0.29}I^{0.37}S^{-0.29})-MTD$$

Gallaway model:(4)$$WFD=0.00338\times \left(\frac{MTD^{0.11}L^{0.43}I^{0.59}}{S^{0.42}}\right)-MTD$$

Empirical PAVDRN model:(5)$$WFD=\frac{0.00073L^{0.519}I^{0.562}MTD^{0.125}}{S^{0.364}}-MTD$$

### 3.4. Evaluation Criteria

MAE, RAE, and RRSE were employed, as defined in Equations (6)–(8), to compare the predictive performance of WFD models in this study.

MAE quantifies the average magnitude of errors between the observed (W*i*) and the estimated values (${\hat{W}}_{i}$). It is calculated as the average of the absolute differences between these values.

RAE serves as an accuracy measure for regression models by comparing the model’s performance to a naive baseline. A well-performing model will yield a RAE ratio of less than 1.

RRSE normalizes the error to the same scale as the original measurements. A robust predictive model will result in error values for MAE, RAE, and RRSE that are close to zero.

These criteria provide a comprehensive assessment of the predictive accuracy and reliability of the models under comparison.(6)$$MAE=\frac{\sum_{i=1}^{x}\left|W_{i}-{\hat{W}}_{i}\right|}{x}$$(7)$$RAE=\frac{\sqrt{\sum_{i=1}^{x}{\left({\hat{W}}_{i}-W_{i}\right)}^{2}}}{\sqrt{\sum_{i=1}^{x}{W_{i}}^{2}}}$$(8)$$RRSE=\sqrt{\frac{\sum_{i=1}^{x}{\left({\hat{W}}_{i}-W_{i}\right)}^{2}}{\sum_{i=1}^{x}{\left(W_{i}-{\bar{W}}_{i}\right)}^{2}}}$$
where $W_{i}$ = actual WFD for case *i*, ${\hat{W}}_{i}$ *=* estimated WFD for case *i*, and ${\bar{W}}_{i}$ = average actual value of WFD for case *i.*

## 4. Experimental Results

### 4.1. Experimental Design

Several limitations in experimental setups were found in previous studies, such as the following:Point-by-point observations using a water level gauge, which may introduce visual measurement errors;Inability to obtain multi-point data simultaneously;Difficulty in achieving uniform rainfall distribution due to the spray configuration and placement.

This study addressed these limitations through a carefully designed experiment focused on achieving uniform rainfall distribution by adjusting the spacing of the spray nozzles. Additionally, using laser measurement and electrical data collection instruments, the experiment allowed for the precise measurement of multiple WFD values at different points simultaneously along the eight-meter-wide runway pavement cross-section.

Figure 7 illustrates the configuration of the water supply system at the rear end. The spray nozzles are positioned 1.8 m above the pavement surface, and the water flow rate is controlled via a shutoff valve beneath the flow meter, adjusting the water flow rate from 25 to 50 L per minute. The water is delivered through hoses to the spray nozzles, corresponding to rainfall intensities of 6.43 mm to 12.87 mm per six minutes. Subsequently, the rainfall intensity recorded every 6 min will be multiplied by 10, representing the hourly rainfall intensity.

During typical aircraft landings, the main landing gear typically lands within 14 m of the runway centerline (the highest point of the runway cross-section), with a lateral distance of 7 m on each side. This study designed a 1.2 m wide, 8 m long asphalt pavement section positioned on a platform 1 m above the ground to conservatively simulate the runway cross-section. The pavement’s transverse slope is adjustable from 0.3% to 1.5%. The upstream end of the pavement represents the runway centerline, and the downstream end is located 8 m from the centerline. Two asphalt concrete pavement samples were fabricated with average mean texture depths of 0.38 mm and 0.98 mm, according to the measurement standards (ASTM E965-15) outlined by the American Society for Testing and Materials [26].

In this experiment, eight spray nozzles were used, each with a maximum spray angle of 120° (Figure 8). Theoretically, the spray distribution should be uniform under the same pressure and water flow conditions. To test the spray coverage, the experiment maintained a water supply at 4 kg/cm^2^ air pressure and a flow rate of 45 L per minute. Table 2 shows the average water output from each nozzle for over 6 min at various nozzle spacings, ranging from 96 to 180 cm. At a nozzle spacing of 120 cm, the average rainfall intensity per 6 min was 13.63 mm with a standard deviation of 1.1 mm and a variance of 0.08, representing the smallest variation among all spacing conditions. Therefore, the nozzle spacing was initially set at 120 cm and adjusted to optimize the uniformity of water distribution.

The pressure and flow loss differ because the distance from the pressurized steel drum to each nozzle varies, as well as the positioning of the branching pipes, necessitating fine-tuning of the nozzle spacing. After a trial-and-error process, the optimal spacing arrangement was selected at 23, 79, 219, 335, 469, 594, 704, and 784 cm from the upstream end. This configuration ensures the most uniform rainfall distribution across the pavement, from the upstream end (representing the runway centerline) to eight meters downstream.

Figure 9 illustrates the laser displacement sensors used for measuring WFD. Six measurement points were established at distances of 100, 190, 310, 440, 590, and 690 cm from the upstream end. A red semiconductor laser with a wavelength of 655 nm projects onto each measurement point, with a response time of five microseconds. On the pavement surface, the laser forms a visible red dot, and the sensor’s receiving component detects the time interval of the reflected waveform. The collected data are transmitted via an RS-422 interface to a programmable logic controller (PLC). WFD measurements are recorded twice per second and continuously logged for 6 min.

### 4.2. Data Collection

Based on the experimental design, this study collected 228 experimental data combinations for WFD measurements (NTU dataset). The combinations included variations in the transverse slope (S) at 0.3%, 0.5%, 1%, and 1.5%; outflow rates of 25, 30, 35, 40, 45, and 50 L per minute, corresponding to rainfall intensities (I) of approximately 64, 77, 90, 102.9, 115.8, and 128.7 mm/h, respectively; mean texture depths (T) of 0.38 mm and 0.98 mm; and drainage length (L) at 100, 190, 310, 440, 590, and 690 cm.

Figure 10 illustrates the measurement results for a transverse slope of 0.5%, mean texture depth of 0.38 mm and a water supply rate of 45 L per minute, collected at six measurement points (L) from the centerline. “O-100” represents the raw measurement data obtained from the laser displacement sensors, while “SMA-100” indicates the smoothed and averaged data after applying the SMA method to the original measurements.

WFD was recorded twice per second, resulting in 720 data points per measurement location over 6 min. Variability in the raw measurements was expected and deemed reasonable due to continuous rainfall on the pavement surface.

During each experiment, the WFD gradually increases before stabilizing as water flows downstream through the measurement points. Measurements were collected from the 300th to the 600th point (i.e., from two minutes thirty seconds to five minutes) of each test to ensure water flow stability. The average of these measurements was recorded as the WFD for the given experimental setup. This process was repeated across all six measurement points, yielding data for each measurement point in every experimental scenario.

### 4.3. Experimental Data Analysis

Table 3 presents the descriptive statistics of the parameters for the NTU dataset from this study and the Gallaway dataset, which originates from prior research (Gallaway et al. [26]). The NTU dataset includes five parameters for each data point: mean texture depth (T), pavement transverse slope (S), rainfall intensity (I), drainage length (L), and water film depth (WFD). The units of measurement are millimeters for parameters T and WFD, centimeters for L, and millimeters per hour for rainfall intensity I. The pavement transverse slope values range from 0.3% to 1.5%, while the drainage length spans from a minimum of 100 cm to a maximum of 690 cm. In this study, WFD is measured as the distance from the water film surface to the highest point of the pavement texture, with the texture peak considered the zero point. As a result, all WFD values are positive, ranging from 0.31 mm to 3.22 mm.

In contrast, the Gallaway dataset comprises 960 data points, with WFD values ranging from −7.26 mm to 4.78 mm. In that dataset, the texture peak serves as the reference plane, with zero indicating the top of the texture. Negative WFD values represent depths below the texture peaks.

Figure 11 illustrates the histograms of WFD values for the NTU and Gallaway datasets, along with their corresponding normal distribution curves. In the NTU dataset, most WFD values fall within the range of 0 to 3 mm, with only about five data points exceeding 3 mm, accounting for approximately 2% of the entire dataset. This suggests that future predictive models may exhibit higher error rates for WFD values above 3 mm. In contrast, the Gallaway dataset shows the majority of WFD values ranging from −1 to 2 mm, including several negative WFD values, with the mean value close to zero.

The NTU dataset was analyzed, as illustrated in Figure 12a. The possible WFD values were determined under varying drainage lengths, transverse slopes, and rainfall intensity conditions from the dataset for a fixed texture depth of 0.38 cm. These values are represented within the I-bar range shown in the figure, yielding an average WFD value of 1.25 cm. Similarly, a fixed texture depth of 0.98 cm, and varying drainage length, transverse slope, and rainfall intensity conditions, resulted in an average WFD value of 1.09 cm. These findings indicate an inverse relationship between higher average texture depth and lower WFD values.

Figure 12b demonstrates a positive correlation between longer drainage lengths and higher average WFD values. Figure 13a reveals that higher pavement transverse slope values are associated with lower WFD values, mirroring the trend observed with average texture depth. In contrast, Figure 13b shows that higher rainfall intensity is positively correlated with higher WFD values, a trend like that observed for texture depth.

Table 4 presents the Pearson correlations for the NTU and Gallaway datasets. For the NTU dataset, at a significance level of α = 0.05, the parameters T, S, I, and L are significantly correlated with WFD. Specifically, T and S exhibit a negative correlation with WFD, whereas I and L show a positive correlation with WFD.

Similarly, for the Gallaway dataset, at α = 0.05, T, S, I, and L are also significantly correlated with WFD. As observed in the NTU dataset, T and S have a negative correlation with WFD, while I and L have a positive correlation. Overall, the correlations between the parameter values and WFD are consistent across the two datasets.

### 4.4. Development of Prediction Model Based on Experimental Data

This study employs MLR to determine the best fit for the data. Using the NTU dataset, the NTU prediction formula (Equations (9) and (10)) was established, yielding an R^2^ value (Coefficient of Determination) of 0.924566.(9)WFD=0.001973×L+0.010448×I−0.547885×S−0.139517×T 
where *L* = drainage length (cm), *I* = rainfall intensity (mm/hr), *S* = pavement transverse slope (%), and *T* = average texture depth (mm).(10)WFD=0.002345×L+0.010448×I−0.021368×S−0.139517×T 
where *L* = drainage length (ft), *I* = rainfall intensity (in/h), *S* = pavement transverse slope (%), and *T* = average texture depth (in).

### 4.5. Comparison of the Performance of Prediction Models

This study compares the performance of four prediction models, the Pourhassan model (Pourhassan et al. [12]), Gallaway model (Gallaway et al. [26]), Empirical PAVDRN model (Huebner et al. [13]), and the NTU model, using the NTU and Gallaway datasets. The comparison utilizes three evaluation metrics: MAE, RAE, and RRSE. Larger values of MAE, RAE, and RRSE indicate poorer WFD prediction performance.

Table 5 and Figure 14 illustrate that, for the NTU dataset, the NTU model provides the most accurate predictions among the five models, with MAE, RAE, and RRSE values of 0.01, 0.59, and 0.10, respectively, indicating superior WFD prediction performance. Table 6 and Figure 15 show that for the Gallaway dataset, the NTU model also achieves the best performance with MAE, RAE, and RRSE values of 0.02, 0.4, and 0.12, respectively. In conclusion, the NTU model demonstrates exceptional performance not only on the NTU dataset but also on the Gallaway dataset. Overall, the comparative analysis highlights the accuracy and robustness of the NTU model among the five evaluated models.

### 4.6. Sensitivity Analysis and Application of NTU Model Variables

Although the NTU model uses four variables to determine the WFD, in practice, airports can only reduce the WFD by adjusting the average texture depth or pavement slope. A sensitivity analysis is conducted to understand the impact of these two variables on the WFD, keeping the rainfall intensity and distance from the runway centerline fixed.

The average texture depths were set at 0.38, 0.58, 0.78, and 0.98 mm, corresponding to a division of 0.38 to 0.98 mm into three equal intervals. Similarly, the runway slope was set at 0.5%, 0.75%, 1%, 1.25%, and 1.5%, with a rainfall intensity of 100 mm per hour and 600 cm from the runway centerline. In each trial, only one variable was altered at a time, with the first setting value for each variable used as the baseline. When modifying one variable, the other three variables were kept at their baseline values, resulting in five WFD values.

For example, when varying the average texture depth, values of 0.58, 0.78, and 0.98 mm were associated with increases of 52%, 105%, and 157%, respectively, compared to the 0.38 mm baseline. The resulting WFD values were reduced by 2.21%, 4.32%, and 6.49%, respectively. Therefore, dividing the percentage change in WFD by the percentage change in texture depth yielded a sensitivity of 0.41. Using the same calculation method, the runway slope was found to have a sensitivity of 0.18, which is lower than that of the average texture depth.

This result suggests airport managers should prioritize increasing the average roughness of the texture, such as removing tire rubber deposits and/or grooving, as the most effective method for reducing the WFD.

### 4.7. Application of the NTU Model at Taiwanese Airports

This study applied the NTU model to three major international airports in Taiwan, referred to as Airports A, B, and C. The rough texture depths of the runway touchdown zones, influenced by tire rubber deposits, were set at 0.25, 0.4, and 0.78 mm, respectively. These measurements were taken at 8 m from the runway centerline, with runway slopes set at 1.4%, 1%, and 1.2%. The WFD threshold for hydroplaning was set at 3 mm (ICAO Annex 14, 2018). Using the NTU model, the critical rainfall intensities for each airport were calculated, yielding values of 212 mm/h, 193 mm/h, and 209 mm/h, respectively.

A comparison with the hourly rainfall data from the past 10 years for these three airports revealed the rainfall was significantly lower than the critical rainfall threshold, indicating that, based on the rainfall data from the past 10 years at these three international airports in Taiwan, the areas within 8 m of the runway centerline are unlikely to reach the 3 mm WFD threshold required for hydroplaning formation. Furthermore, a review of aviation incidents involving heavy rainfall at these airports showed no cases in which aircraft encountered a water film depth exceeding 3 mm upon landing, which would have led to hydroplaning.

## 5. Discussion and Conclusions

This study developed the NTU prediction model to forecast water film depth under various environmental conditions, providing a more reasonable and feasible predictive model and subsequent early warning measures for airports to assess whether the water film depth is sufficiently deep to endanger aircraft safety.

Water film depth on the pavement surface is one of the primary factors contributing to hydroplaning incidents. This study designed experimental equipment specifically for runway conditions to accurately predict WFD in real time. The custom-designed rainfall nozzles and adjustable positioning of the equipment ensured a uniform rainfall intensity across every point on the runway. Additionally, this study employed a laser displacement sensor and a programmable logic controller to obtain high-accuracy, high-sampling-rate WFD data. Modern automated data acquisition enables simultaneous measurement at multiple points and captures the complete WFD curve from zero to a stable depth, which was previously difficult to obtain.

Four commonly applied parameters, including mean texture depth, runway transverse slope, rainfall intensity, drainage length, and water film depth, were used to establish the NTU dataset. A multivariable linear regression approach was employed to build the predictive model (NTU model). The NTU and the Gallaway datasets were used to compare four models: NTU, POURHASSAN, GALLAWAY, and Empirical PAVDRN. The models were evaluated using MAE, RAE, and RRSE. The results indicated the NTU models exhibited lower prediction errors compared to the other models. Specifically, the NTU model not only performed well with the NTU dataset but also showed excellent performance with the Gallaway dataset, with the predicted water film depths being closer to the actual water film depths compared to other models.

This study uses laser displacement sensors to measure WFD based on reflections from texture peaks that allow a better estimation of the actual water film thickness. However, most empirical models calculate WFD from the reference plane. Using laser displacement sensors is relatively new and the measurements obtained from texture peak and reference plane are different; therefore, it is quite possible that the differences may affect model predictions and comparisons. It is suggested that future research can build on this study to develop more objective models for further analysis.

This study also found that among the controllable parameters affecting water film depth, the average texture depth of the pavement had a significant impact and should be a focus for airport management. Furthermore, when applying the NTU model to the runways of three international airports in Taiwan, it was found that, based on the hourly rainfall data from the past 10 years at these airports, the areas within 8 m of the runway centerline did not reach the 3 mm critical water film depth threshold. However, if the hourly rainfall exceeds 190 mm, special attention should be given to prevent the water film depth from reaching the critical threshold. However, this study did not consider the effect of wind on water film depth. Future experiments could further explore the effect of wind on water film depth.

## Figures and Tables

**Figure 1 sensors-25-02202-f001:**
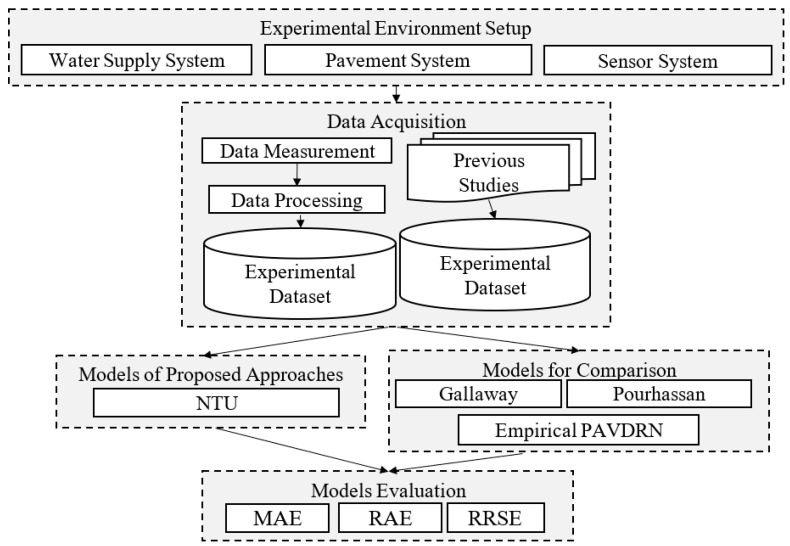
The experimental and evaluation process for developing the WFD prediction model.

**Figure 2 sensors-25-02202-f002:**
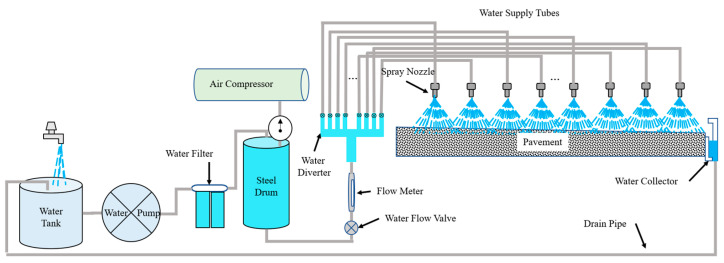
Layout of water supply system.

**Figure 3 sensors-25-02202-f003:**
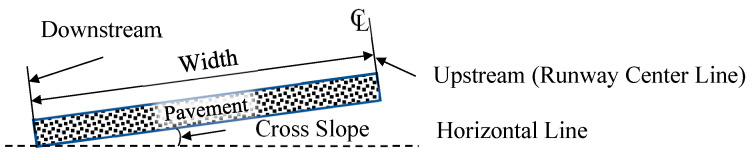
Runway transverse section.

**Figure 4 sensors-25-02202-f004:**
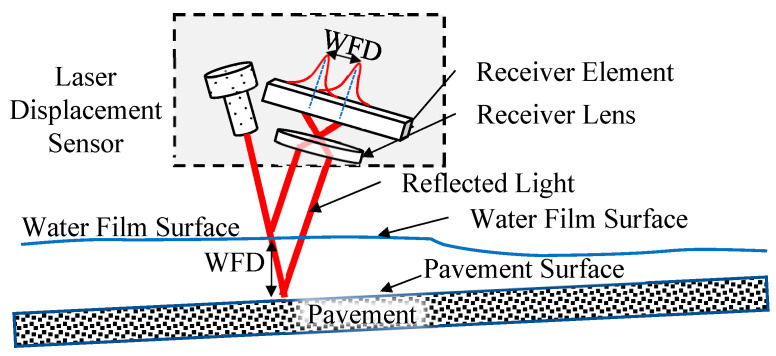
Laser displacement sensor.

**Figure 5 sensors-25-02202-f005:**
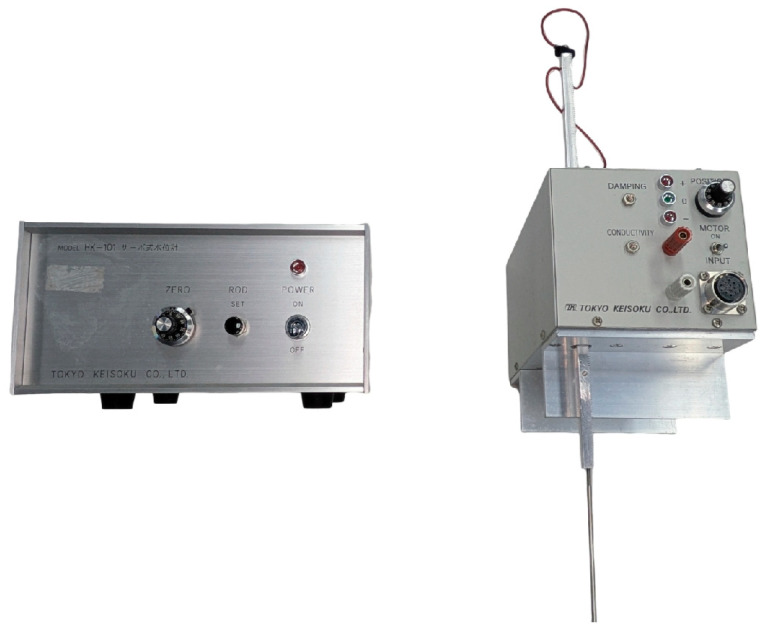
Probe-type water level gauge.

**Figure 6 sensors-25-02202-f006:**
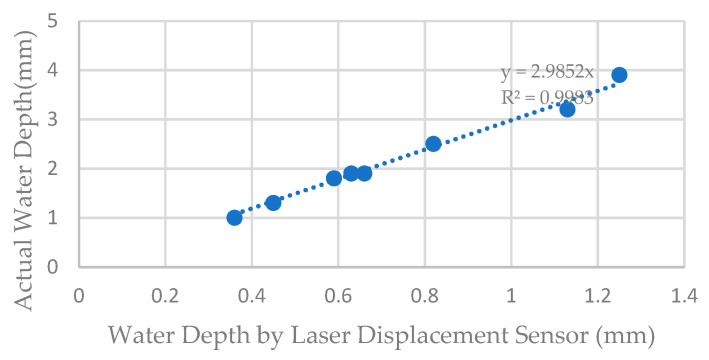
The relationship between actual water depth and that measured by laser displacement sensor.

**Figure 7 sensors-25-02202-f007:**
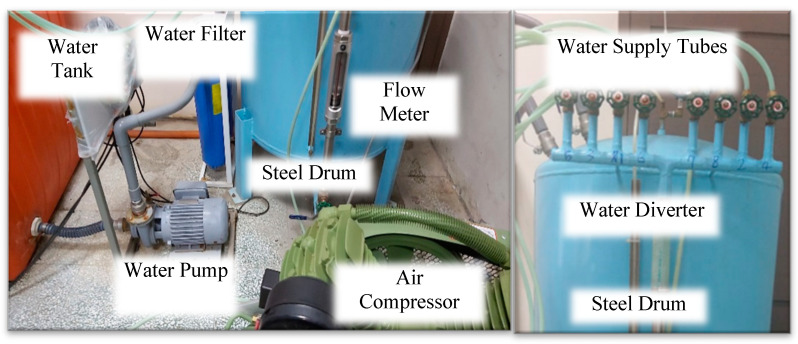
Front-end of the water supply system.

**Figure 8 sensors-25-02202-f008:**
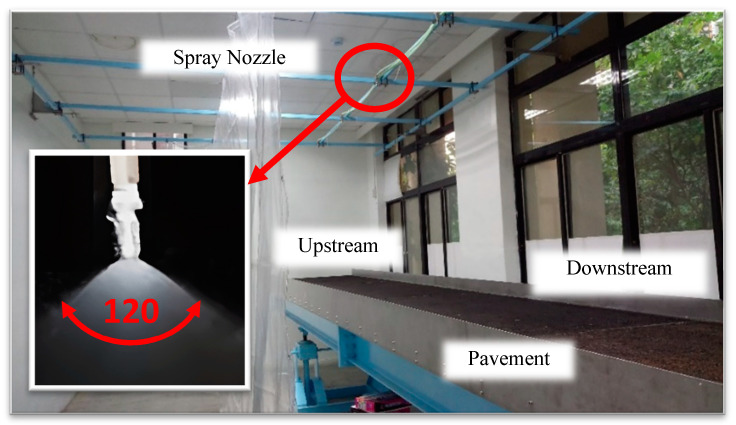
Back-end of water supply system with spray nozzles.

**Figure 9 sensors-25-02202-f009:**
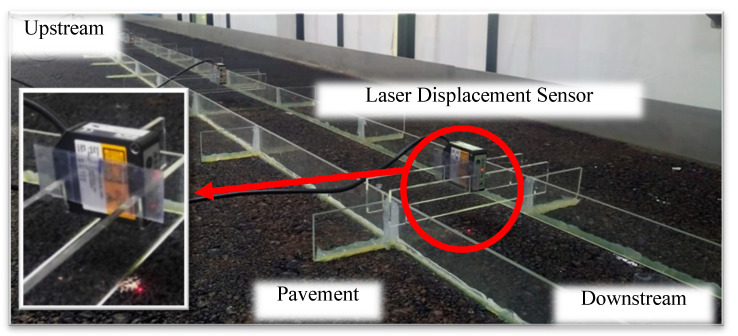
Pavement with laser displacement sensors.

**Figure 10 sensors-25-02202-f010:**
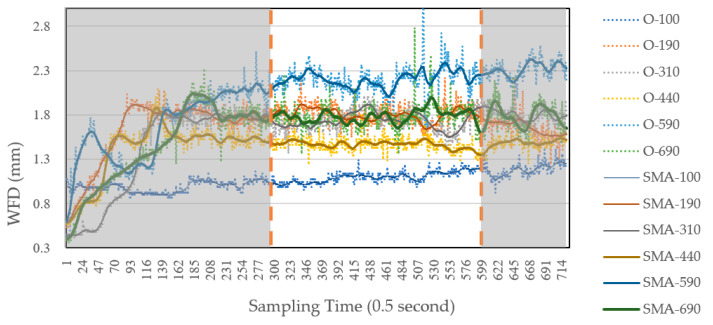
The table of raw WFD measurements and SMA-processed values under the conditions of a mean texture depth of 0.38 mm, a transverse slope of 0.5%, and an outflow rate of 45 L per minute.

**Figure 11 sensors-25-02202-f011:**
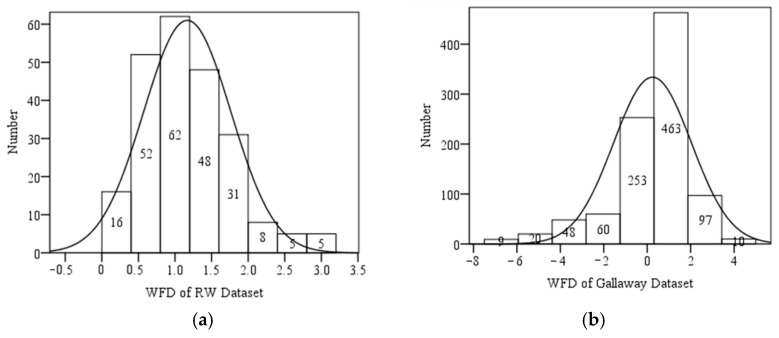
(**a**) Histogram of NTU dataset; (**b**) histogram of Gallaway dataset.

**Figure 12 sensors-25-02202-f012:**
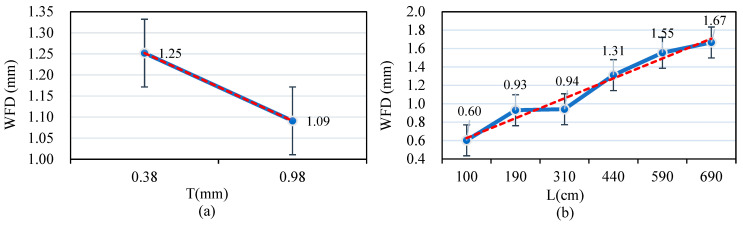
(**a**) Correlations between WFD and texture depth; (**b**) correlations between WFD and drainage length.

**Figure 13 sensors-25-02202-f013:**
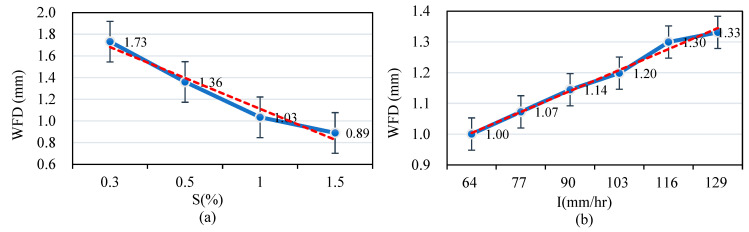
(**a**) Correlations between WFD and transverse slope; (**b**) correlations between WFD and rainfall intensity.

**Figure 14 sensors-25-02202-f014:**
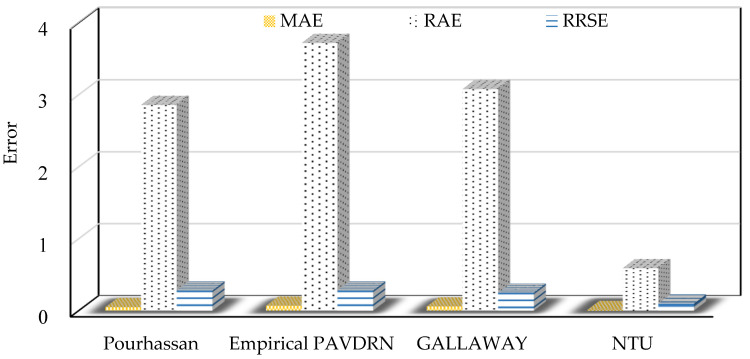
Compare the MAE, RAE, and RRSE error values of various prediction models using the NTU dataset.

**Figure 15 sensors-25-02202-f015:**
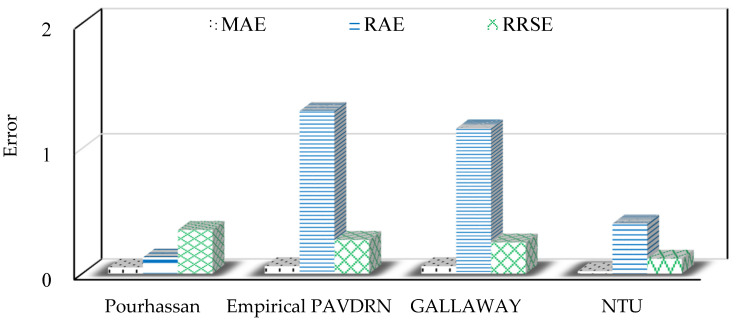
Compare the MAE, RAE, and RRSE error values of various prediction models using the Gallaway dataset.

**Table 1 sensors-25-02202-t001:** Comparison of detection equipment.

Sensors	Principle	Precision	Advantages/Disadvantages
DRS511	Electrical	0.1 mm	Measuring water film depth/difficult to replace and maintain; subjected to repeated action of environment and load; short life cycle.
DSC111	Camera	0.01 mm	Monitoring thin water film with low error/unable to monitor water film depth of more than 2 mm; weather affection.
HKT-10	Electrode	0.1 mm	Need to contact the measured water surface; must contain electrolyte in water.
CD33-50N	Laser-Based	5 μm	High accuracy; small size/cannot be too far away from the measured water surface.
Water Depth Gauge_NASA	Visual	0.01 in	The error is too large; measuring a point one time.

**Table 2 sensors-25-02202-t002:** Rainfall per 6 min within different spacings of spray nozzles.

Spacing of Spray Nozzle (cm)	Mean (mm)	Std. Dev. (mm)	COV
96	15.55	3.15	0.20
108	14.71	1.48	0.10
120	13.63	1.10	0.08
132	12.55	1.36	0.11
144	11.64	1.33	0.11
156	10.74	13.29	1.23
168	9.88	10.04	1.02
180	9.19	8.88	0.97

**Table 3 sensors-25-02202-t003:** Descriptive statistics for feature values.

Dataset	Feature	Min.	Max.	Mean	Std. Error	Median	Std. Deviation	Variance	Range
NTU	*T* (mm)	0.38	0.98	0.70	0.02	0.98	0.30	0.09	0.60
	*S* (%)	0.30	1.50	0.94	0.03	1.00	0.45	0.21	1.20
	*I* (mm/hr)	64.33	128.65	95.13	1.47	90.06	22.24	494.68	64.33
	*L* (cm)	100.00	690.00	386.67	13.89	375.00	209.80	44,015.27	590.00
	*WFD* (mm)	0.31	3.22	1.17	0.04	1.08	0.60	0.36	2.91
Gallaway	*T* (mm)	0.08	4.17	1.34	0.05	0.89	0.91	1.43	4.09
	*S* (%)	0.50	8.00	2.47	0.06	2.00	0.50	1.89	7.50
	*I* (mm/hr)	5.59	161.54	61.17	1.45	53.09	12.45	45.07	155.96
	*L* (cm)	152.40	737.62	460.53	6.55	467.87	182.88	202.80	585.22
	*WFD* (mm)	−7.26	4.78	0.24	0.06	0.53	0.46	1.79	12.04

**Table 4 sensors-25-02202-t004:** Pearson correlation for feature values.

Dataset	Feature	*T* (mm)	*S* (%)	*I* (mm/hr)	*L* (cm)	WFD (mm)
NTU	*T* (mm)	1.00	−0.12	−0.06	0.00	−0.14 *
	*S* (%)	−0.12	1.00	0.06	0.00	−0.45 **
	*I* (mm/h)	−0.06	0.06	1.00	0.00	0.20 **
	*L* (cm)	0.00	0.00	0.00	1.00	0.62 **
	WFD (mm)	−0.14 *	−0.47 **	0.20 **	0.062 **	1.00
Gallaway	*T* (mm)	1.00	0.05	−0.04	−0.01	−0.71 **
	*S* (%)	0.05	1.00	0.01	−0.00	−0.33 **
	*I* (mm/h)	−0.04	0.01	1.00	0.00	0.46 **
	*L* (cm)	−0.01	−0.00	0.00	1.00	0.24 **
	WFD (mm)	−0.71 **	−0.33 **	0.46 **	0.24 **	1.00

* Correlation is significant at the 0.05 level (two-tailed). ** Correlation is significant at the 0.01 level (two-tailed).

**Table 5 sensors-25-02202-t005:** Compare the MAE, RAE, and RRSE error values of various prediction models using the NTU dataset.

Approach	Pourhassan	Empirical PAVDRN	Gallaway	NTU
MAE	0.05	0.07	0.06	0.01
RAE	2.86	3.72	3.08	0.59
RRSE	0.28	0.28	0.25	0.10

**Table 6 sensors-25-02202-t006:** Compare the MAE, RAE, and RRSE error values of various prediction models using the Gallaway dataset.

	Pourhassan	Empirical PAVDRN	Gallaway	NTU
MAE	0.05	0.06	0.06	0.02
RAE	0.13	1.30	1.16	0.40
RRSE	0.35	0.27	0.25	0.12

## Data Availability

The data presented in this study are available on request from the author.

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
