# Peer review of "Development of a Predictive Model for Runway Water Film Depth"

_sensors, 2025, doi:10.3390/s25072202_

Round 1
Reviewer 1 Report
Comments and Suggestions for Authors
In this paper, a WFD dataset specific to different runway conditions was developed with measuring four parameters, rainfall intensity, pavement mean texture depth, drainage length, and transverse slope. Multiple linear regression method is employed to establish a model for WFD predictions compared with four existing empirical models using NTU and Gallaway datasets. However, the paper exists too much problems and comments are as follows,
- In the Section of “Abstract”, more research findings should be contained with specific data and innovation points should be point out.
- In section of 2.2, There are many models for predicting water film thickness, such as Gallaway Model (1971), NASA Model (1974), French Model (SETRA, 1994), et al. It is suggested to summarize and compare their advantages and disadvantages in a table.
- In section of 2.4, the overview of the water film thickness detection equipment is not comprehensive. Various instruments and techniques are employed to measure WFD, including Laser-Based Sensors, Ultrasonic Sensors, Camera-Based Systems, et al. Thus, it is suggested that a table be used to summarize all the contact and non-contact testing equipment.
- In this research, the predictability of the proposed National Taiwan University (NTU) model is compared with four existing empirical models using NTU and Gallaway datasets. Generally, it is recommended to validate a prediction model using two or more empirical models to ensure its accuracy and reliability. Therefore, it is suggested that the proposed National Taiwan University (NTU) model is better to be compared with Gallaway datasets, NASA Model or French Model.
- The Summary and Findings part needs to be further refined and highlight the innovation point, indicate the specific research results.
- The English writing of the manuscript should be improved by a professional English editor, and the grammar need to be checked seriously.
The English writing of the manuscript should be improved by a professional English editor, and the grammar need to be checked seriously.
Author Response
Comments 1:In the Section of “Abstract”, more research findings should be contained with specific data and innovation points should be point out.
Responses1:The authors sincerely appreciate the reviewer’s feedback and has added the specific data and key innovation points to Abstract (Row 22): This study employed a Laser Displacement Sensor and a programmable logic controller to obtain high-accuracy, high-sampling-rate WFD data. Modern automated data acquisition enables simultaneous measurement at multiple points and captures the complete WFD curve from zero to a stable depth, which was previously difficult to obtain.
Comments 2:In section of 2.2, There are many models for predicting water film thickness, such as Gallaway Model (1971), NASA Model (1974), French Model (SETRA, 1994), et al. It is suggested to summarize and compare their advantages and disadvantages in a table.
Responses 2:The authors sincerely appreciate the reviewer's suggestions and added a comparative description of three models in Row 285. But we attempted to locate the NASA and French formulas you mentioned to include in Section 2.2 (Literature Review). However, due to time constraints, we were unable to include those two model’s comparisons in this study. Thank you for your understanding.
Comments 3:In section of 2.4, the overview of the water film thickness detection equipment is not comprehensive. Various instruments and techniques are employed to measure WFD, including Laser-Based Sensors, Ultrasonic Sensors, Camera-Based Systems, et al. Thus, it is suggested that a table be used to summarize all the contact and non-contact testing equipment.
Responses 3:
The authors sincerely appreciate the reviewer's suggestions and fully agree with the reviewer's perspective. Therefore, in Section 2.4, the contact and non-contact detection equipment has been compiled and added to Row 171:
The sensors measure WFD using different principles, including Laser-Based Sensors, Ultrasonic Sensors, and Camera-Based Systems, which can generally be categorized as contact and non-contact types. The author has referenced the sensor data compiled by Ling [1] for WFD measurement and incorporated the two sensors used in this study, along with the NASA’s water depth gauge into Table 1.
Table 1. Comparison of Detection Equipment
|
Sensors |
Principle |
Precision |
Advantages/Disadvantages |
|
DRS511 |
Electrical
|
0.1mm |
Measuring water-film depth/difficult to replace and maintain; Subjected to repeated action of environment and load; Short life cycle. |
|
DSC111 |
Camera |
0.01mm |
Monitoring thin water film with low error/Unable to monitor water-film depth of more than 2 mm; Weather affection. |
|
HKT-10 |
Electrode |
0.1mm |
Need to contact the measured water surface; must contain electrolyte in water. |
|
CD33-50N |
Laser-Based |
5μm |
High accuracy; small size/cannot be too far away from the measured water surface. |
|
NASA Water Depth Gauge |
Visual |
0.01 in |
The error is too large; Measuring a point one time. |
Comments 4:In this research, the predictability of the proposed National Taiwan University (NTU) model is compared with four existing empirical models using NTU and Gallaway datasets. Generally, it is recommended to validate a prediction model using two or more empirical models to ensure its accuracy and reliability. Therefore, it is suggested that the proposed National Taiwan University (NTU) model is better to be compared with Gallaway datasets, NASA Model or French Model.
Responses 4:
The authors sincerely appreciate the reviewer’s suggestions. Additional explanations regarding the limitations of the comparative models and evaluation criteria have been added. The revision is reflected in Row 192 of Chapter 3 of the paper as follows:
Regarding the WFD prediction model, an empirical model proposed by Gallaway et al. has already been introduced. Additionally, various models with different levels of accuracy and reliability evaluation criteria have been adopted. Since using a combination of different comparison and evaluation approaches presents a significant challenge, this study focuses on exploring and analyzing commonly used empirical models and evaluation criteria.
Comments 5:The Summary and Findings part needs to be further refined and highlight the innovation point, indicate the specific research results.
Rsponses 5:The authors thank your comments and have refined and highlighted the innovation points in Chapter 5, Paragraph 2 (Row546) as follows:…this study employed a Laser Displacement Sensor and a programmable logic controller to obtain high-accuracy, high-sampling-rate WFD data. Modern automated data acquisition enables simultaneous measurement at multiple points and captures the complete WFD curve from zero to a stable depth, which was previously difficult to obtain.
Comment 6:The English writing of the manuscript should be improved by a professional English editor, and the grammar need to be checked seriously.
Responses6:Thank you for your comments. The entire manuscript has been professionally edited once again and underwent a thorough grammar check.
Reviewer 2 Report
Comments and Suggestions for Authors
The system developed in the manuscript has the advantage of controllable conditions compared to on-site sensor testing. However, from the perspective of sensors, it seems necessary to discuss the accuracy and stability of this laser sensor in the presence of water.
Author Response
Comment1:The system developed in the manuscript has the advantage of controllable conditions compared to on-site sensor testing. However, from the perspective of sensors, it seems necessary to discuss the accuracy and stability of this laser sensor in the presence of water.
Response1:The authors sincerely appreciate the reviewer's comments. Therefore, in Section 2.4, an explanation of the accuracy of the Laser Displacement Sensor (CD33-50N) has been added in Table 1(Row176), and a chart for the conversion formulas has been included in Section 3.1.3 as follows:Row 244 has been updated with the following text:
Before the experiment, the Probe-Type Water Level Gauge and Laser Displacement Sensor were used to measure water depth, obtaining the correlation: Actual water depth = 2.9852 * Laser Displacement Sensor measured water depth, as shown in Figure 6.
Fig 6. The relationship between actual water depth and measured by Laser Displacement Sensor.
Reviewer 3 Report
Comments and Suggestions for Authors
The paper presents an experimental study performed with the objective to develop improved model for water film depth for airport runways.
The experimental study is well developed. However, there are several issues regarding validation of the model and comparison to other available models.
The model claimed to be PAVDRN model (equation 6) is actually Gallaway model, using imperial system of units, and the constant is wrong. The actual PAVDRN model is provided in the NCHRP report: Transportation Research Board. 1998. Improved Surface Drainage of Pavements: Final Report. Washington, DC: The National Academies Press. https://doi.org/10.17226/6357, page 51, equation 10.
In addition, the model presented in Equation 7 is not provided in reference 11, as stated in the paper.
The models presented in equations 4 to 7, including the correct PAVDRN model, use reference plane for determination of WFD to be equal to MTD, while the proposed model uses texture peak. It is necessary to explain how this difference is taken into consideration when comparing models.
The models presented in equations 4 to 7, including correct PAVDRN model, use different units for same parameters and it is necessary to specify units used for each model. It is question if authors used correct units for each model when comparing them, because the MAE, RAE and RRSE values seem to be unrealistically large for these models.
Some more specific comments:
- Row 11: Should be “accident occurrences” instead of “aviation occurrences”
- References No. 9 and No. 15 are identical
Author Response
Comments1:The model claimed to be PAVDRN model (equation 6) is actually Gallaway model, using imperial system of units, and the constant is wrong. The actual PAVDRN model is provided in the NCHRP report: Transportation Research Board. 1998. Improved Surface Drainage of Pavements: Final Report. Washington, DC: The National Academies Press. https://doi.org/10.17226/6357, page 51, equation 10.
Response1:The author sincerely appreciates the reviewer's comments. The equation mentioned by the reviewer refers to the Analytical PAVDRN model, not the Empirical PAVDRN model presented in this paper. It has been confirmed that the constants of Empirical PAVDRN formula in this paper are correct, but the formula description does indeed require some revisions, as follows:
- The unit in Row 283~285 has been corrected to imperial units.
- The constant of Gallaway formula in Row 293 has been corrected to imperial units.
- The name of “PAVDRN model“ in Row 294 has been modified to be “Empirical PAVDRN model”.
Comments2: In addition, the model presented in Equation 7 is not provided in reference 11, as stated in the paper.
Response2: The authors sincerely appreciate the reviewer's correction. The authors made the revision and Equation 7 has been removed from the paper.
Comments3: The models presented in equations 4 to 7, including the correct PAVDRN model, use reference plane for determination of WFD to be equal to MTD, while the proposed model uses texture peak. It is necessary to explain how this difference is taken into consideration when comparing models.
Response3: The authors sincerely appreciate the reviewer's reminder and corrections. This study uses Laser Displacement Sensors to measure WFD based on reflections from texture peaks that allows better estimation about the actual water film thickness. However, most empirical models calculate WFD from the reference plane. Since using laser displacement sensor is relatively new and the measurements obtained from texture peak and reference plane are different; therefore, it is quite possible that the differences may affect model predictions and comparisons. It is suggested that future research can build on this study to develop more objective models for further analysis. A supplementary explanation of the study’s limitations has been revised in Chapter 5, Paragraph 4 (Row561).
Comments4: The models presented in equations 4 to 7, including correct PAVDRN model, use different units for same parameters and it is necessary to specify units used for each model. It is question if authors used correct units for each model when comparing them, because the MAE, RAE and RRSE values seem to be unrealistically large for these models.
Response4: The authors sincerely appreciate the reviewer's correction. All formula units have been rechecked and confirmed to be in imperial units. The MAE, RAE, and RRSE values for the comparative models have also been reviewed and verified as correct.
Comments5:Row 11: Should be “accident occurrences” instead of “aviation occurrences”References No. 9 and No. 15 are identical
Response 5: The authors sincerely appreciate the reviewer's correction and have revised Row 11 as suggested.
Reviewer 4 Report
Comments and Suggestions for Authors
The Manuscript "Development of a Predictive Model for Runway Water Film Depth" is well written. It is suitable for publication. I have no comments.
Author Response
Comments1:The Manuscript "Development of a Predictive Model for Runway Water Film Depth" is well written. It is suitable for publication. I have no comments.
Response1:
The authors sincerely appreciate the reviewer's encouragement. Thank you very much.